# Antimicrobial resistance trend of bacterial uropathogens at the university of Gondar comprehensive specialized hospital, northwest Ethiopia: A 10 years retrospective study

**Desie Kasew**[1]*, **Blen Desalegn**[2], **Mihret Aynalem**[2], **Sosina Tila**[2], **Dureti Diriba**[2], **Beimnet Afework**[2], **Michael Getie**[3], **Sirak Biset**[1], **Habtamu Wondifraw Baynes**[4]

1 Department of Medical Microbiology, School of Biomedical and Laboratory Science, College of Medicine and Health Sciences, University of Gondar, Gondar, Ethiopia, 2 Department of Medical Laboratory Science, School of Biomedical and Laboratory Science, College of Medicine and Health Sciences University of Gondar, Gondar, Ethiopia, 3 Microbiology Laboratory, University of Gondar Comprehensive Specialized Hospital, Gondar, Ethiopia, 4 Department of Clinical Chemistry, School of Biomedical and Laboratory Science, College of Medicine and Health Sciences, University of Gondar, Gondar, Ethiopia

* dessalegnkassaw83@gmail.com, desie.kasew@uog.edu.et

**Data Availability Statement:** Data cannot be shared publicly because of ethical restriction of human research data as the patient data is private

## Abstract

Urinary tract infection and antimicrobial resistance remains the major problem, with significant health and socioeconomic burden, particularly in developing countries. This infection is commonly caused by Gram-negative bacteria, principally by *Escherichia coli*. So, this study aimed to determine bacterial isolates and antimicrobial resistance trend among patients with urinary tract infection at the University of Gondar Comprehensive Specialized Hospital, Northwest Ethiopia. A retrospective study was conducted from January 1st to February 28th. A ten years (2010–2019) record of urine culture results, the biochemical test and antimicrobial susceptibility test results of isolates were collected from the medical microbiology laboratory register using a checklist. Data quality was checked, entered, and analyzed using SPSS version 23. We have presented results through descriptive tables and graphs. The overall prevalence of urinary tract infection among 4441 patients was 24.1%. *Escherichia coli* (37.7%), *Klebsiella pneumoniae* (11.4%), and *Staphylococcus aureus* (9.1%) were the predominant uropathogens. The infection rate was nearly similar across both sexes but highest in the age group above 60 years. Above 75% of Gram-negative isolates were resistant to ampicillin (92.5%), amoxicillin-clavulanate (80.1%), tetracycline (79.3%), cefuroxime (79.2%), and Trimethoprim-sulfamethoxazole (78.3%). Over 2/3 of Gram-positive isolates also showed increased resistance to tetracycline (84.8%) and penicillin (71.6%). Moreover, more than 44% of the isolates were multidrug-resistant (MDR). We have seen an inconsistent trend of antimicrobial resistance, with an overall resistance rate of above 50%. In conclusion, the overall prevalence of urinary tract infection was high and elderly patients were most affected. More than 70% of both Gram positive and gram-negative isolates were resistant to penicillin, ampicillin, amoxicillin-clavulanate, tetracycline, cefuroxime, Trimethoprim-sulfamethoxazole. Above than 44% of the isolates were multidrug-resistant (MDR). The

and the ethical review committee (ERC) of School Biomedical and Laboratory Sciences, College of Medicine and Health sciences, University of Gondar limits publicly sharing the data. Data are available from the ethical review committee (ERC) (contact via the E-mail: abty12@gmail.com or deme2112@gmail.com) for researchers who meet the criteria for access to confidential data. The data underlying the results presented in the study are available from the Ethical Review Committee (ERC) of School Biomedical and Laboratory Sciences, College of Medicine and Health sciences, University of Gondar can be accessed by the E-mail abty12@gmail.com or deme2112@gmail.com.

**Funding:** The funders had no role in study design, data collection and analysis, decision to publish, or preparation of the manuscript. The authors received no specific funding for this work.

**Competing interests:** The authors have declared that no competing interests exist.

**Abbreviations:** AMR, Antimicrobial resistance; AST, Antimicrobial Susceptibility Test; CLED, Cysteine Lactose Electrolyte Deficient; CLSI, Clinical Laboratory Standards Institute; CMHS, Collage of Medicine and Health Science; MDR, Multidrug Resistance; MRSA, Methicillin resistant *staphylococcus aureus*; SBLS, School of Biomedical and Laboratory Science; UoGCSH, University of Gondar Comprehensive Specialized Hospital; UTI, Urinary tract infection.

increasing rate of antimicrobial resistance calls for routine diagnosis and antimicrobial susceptibility testing. A prospective multicenter study indicating the status of resistance should be encouraged.

## Introduction

Urinary tract infection (UTI) is one of the most common infectious diseases, particularly in developing countries, overwhelmed with healthcare and economic constraints [1]. Urinary tract infection can be called pyelonephritis (kidney infection), or cystitis (bladder infection). The infection has clinical signs and symptoms such as dysuria, frequency, urgency, suprapubic tenderness, fever, chills, nausea, and vomiting [2, 3]. The bacterial causes of UTI include *Escherichia coli (E. coli)* (which causes 80% of the UTI), *Klebsiella pneumoniae (K. pneumoniae)*, *Citrobacter* species, *Enterobacter* species, *Pseudomonas aeruginosa (P. aeruginosa)*, and *Staphylococcus* species [4–6]. The mechanism of pathogenesis of the mentioned pathogens include adhesion to the host cell epithelium, invasion, immune evasion via cell wall lipopolysaccharide, capsule, and fimbriae [7]. The infection is higher in females due to biological factors such as the short urethra, anal-genital proximity, and use of spermicides [8].

Urinary tract infection is associated with increased resistance to antimicrobial agents such as multidrug resistance (MDR) with substantial medical and a financial burden [9, 10]. Antimicrobial resistance is a thoughtful medical problem in which microorganisms use varied resistance mechanisms such as horizontal gene transfer (such as plasmids and bacteriophages), genetic recombination, and mutations [11]. In addition, self-medication [12], empirical therapy, misuse, and overuse of antimicrobials which are highly practiced in Ethiopia, hasten antimicrobial resistance (AMR) end up in prolonged illness, disability, increased health care costs, and death [1, 13, 14]. In the era of rising antimicrobial resistance, current longitudinal studies revealing the prevalence and AMR trend of uropathogens are crucial to coming up with this problem [15]. This up-to-date evidence will support clinicians to identify the etiology of UTI, ensure appropriate empirical treatment for a reasonable period and an affordable cost. Moreover, it helps health policymakers in implementing locally efficient therapy and preventive guidelines. Although there are snapshot studies on the prevalence of UTI and associated AMR, data showing results of longitudinal studies lacked in the study area. Hence, this study aimed to assess the prevalence and AMR trend of bacterial uropathogens over 10 years between 2010 and 2019 among patients with UTI at the University of Gondar Comprehensive Specialized Hospital (UoGCSH), Northwest Ethiopia.

## Materials and methods

### Study area, design and period

A Hospital-based retrospective study was conducted by retrieving laboratory record of ten years (2010–2019) data from January 1st to February 28th, 2020 at the UoGCSH, Gondar, Ethiopia. The University hospital is one of the pioneering tertiary level referral and teaching hospitals in the country, which serves more than 5 million people in Gondar province and neighboring regions. It has different service centers in inpatient and outpatient settings such as fistula, cancer, dialysis, psychiatric and ophthalmology clinics. It also has an organized laboratory such as microbiology and mycobacteriology sections [16]. We have collected manually, a ten years (2010–2019) retrospective data from the microbiology laboratory logbook which is

a paper-based record of the laboratory results. We have collected data complete record of the variables mentioned in the exclusion criteria.

All recorded urine culture results of patients who visited the UoGCSH and were suspected of UTI were the source. Moreover, the recorded urine culture results of those UTI suspected patients who visited the hospital from 2010–2019 were the study population and analyzed.

Inclusion and exclusion criteria: We have included the records of patients' data which contains the patients' age, sex, urine culture result including antimicrobial susceptibility test (AST) results for significant bacteriuria ($10^5$ CFU/ml) of monomorphic organisms which have been processed and recorded from 2010 to 2019. However, records lacking at least one of the variables age, sex, urine culture results, and AST results of cultures with significant bacteriuria were excluded.

Ethical approval letter was obtained from ethical review committee (ERC) of school of Biomedical and Laboratory Sciences, College of medicine and health sciences, University of Gondar. We explained the study objectives to the heads of the hospital director and laboratory personnel who worked in the hospital. Consent from patients was not obtained as a waiver of consent by the ERC. In addition, we extracted our research data from a record in which patients' name was anonymous.

## Data collection and analysis

We have collected variables such as age, sex, urine culture result, isolated uropathogens, and their AST results from the UoGCSH Microbiology laboratory record book by using a data collection checklist. The urine specimen was first collected with sterile wide mouthed cup and inoculated on Cysteine-Lactose-Electrolyte-Deficient (CLED) agar. Colonies with a significant number ($10^5$ CFU/ml) from CLED agar were subjected to Gram staining and then sub-cultured on MacConkey (Gram negative) and Blood agar plates (gram positive) (BIO MARK Laboratories, India) for identification. Cultures were incubated at $37^0$c for 24 hours. After a series of biochemical tests were performed to identify Gram-negative isolates (performed using Triple sugar iron agar, Urea agar, Citrate agar, Lysine iron agar, Motility medium and Indole test) and Gram-positive isolates (Catalase, coagulase, bile-esculin hydrolysis, and optochin sensitivity). Then, the Kirby-Bauer disk-diffusion method of AST commences on Muller-Hinton agar (BIO MARK Laboratories, India) to determine its susceptibility to antimicrobial agents by incubating at $37^0$c for 18 hours. Antimicrobial discs used were ampicillin (10 μg), amoxicillin-clavulanic acid (20/10 μg), cefoxitin (30 μg), ciprofloxacin (5 μg), gentamycin (10 μg), nitrofurantoin (300 μg), norfloxacin (10 μg), amikacin (30 μg), kanamycin (30 μg), tetracycline (30 μg), tobramycin(10 μg), ceftriaxone (30 μg), nalidixic acid (30 μg), cefuroxime (30 μg), cefotaxime (30 μg), ceftazidime (30 μg), vancomycin(30 μg), meropenem (10 μg), Trimethoprim-Sulfamethoxazole (1.25/23.75 μg), chloramphenicol (30 μg), and penicillin (10 units) (HI Media Laboratories, India). The AST results were collected, and multidrug-resistant (MDR) isolates were identified. Multidrug resistance is the in vitro non-susceptibility to at least one drug in more than two classes of antimicrobial agents [17]. The antimicrobial discs were selected, and AST results were interpreted, based on the clinical laboratory standards institute (CLSI) guideline [18].

The data were summarized and entered into a statistical package for social sciences (SPSS) version 23 software and were analyzed using the software (SPSS) for descriptive statistics. Then, the descriptive results were presented with tables and graphs. The trend of antimicrobial resistance was determined by dividing the number of resistant isolates to the total isolates tested in each year. The data were collected by investigators with data quality and completeness checks throughout the collection period, at the end of data collection, and after entry to SPSS for statistical analysis.

## Results

### Socio-demographic characteristics and rate of infection

From Jan 2010–2019, the UoGCSH bacteriology laboratory analyzed 4441 urine samples from UTI suspected patients. Of those patients, 54.8% were females. The age group 21–30 years accounts for the highest proportion (27.1%) of UTI suspected patients, while the highest prevalence of UTI (47.4%) falls in the age group 61–70 years, and the least affected (15.4%) falls in the age group 2–10 years (**Table 1**).

### The proportion of uropathogenic bacterial isolates

A total of 1072 (24.1%) significant bacteriuria of monomorphic bacterial growth was recorded. Of these isolates, 879 (82%) were Gram-negative bacteria. *Escherichia coli* (37.7%) and *K. pneumoniae* (11.4%) were the predominant of all isolates while *S. aureus* (9.14%) was the leading Gram-positive and the third most common of all uropathogenic isolates in this study (**Table 2**).

### Antimicrobial resistance pattern of Gram-positive isolates

Gram-positive isolates showed a high resistance to tetracycline (84.8%) and penicillin (71.6%). Antimicrobial agents most effective against Gram-positive uropathogens were vancomycin and nitrofurantoin. *Staphylococcus aureus* was the most common isolate comprising about 51% of Gram-positive isolates. It was highly resistant to tetracycline (85.7%) and trimethoprim-sulfamethoxazole (83.5%). In addition, 30% of *S. aureus* were resistant to cefoxitin (**Table 3**).

### Antimicrobial resistance pattern of Gram-negative isolates

Gram-negative isolates showed a high resistance rate to ampicillin (92.5%), amoxicillin-clavulanate (80.1%), tetracycline (79.3%), cefuroxime (79.2%) and trimethoprim-sulfamethoxazole (78.3%). Their resistance against cephalosporin drugs ranges from 50%-79.2%, while fluoroquinolone resistance was 51.5% - 66.5%. The least resistance was reported against amikacin (20%) and meropenem (26.4%). *Escherichia coli*, which accounted for 45.9% of Gram-negative

**Table 1. The distribution of uropathogenic isolates with sex and age at the university of Gondar comprehensive specialized hospital, 2010–2019.**

| Variable | Category of variable | Frequency (%) | UTI (significant bacteriuria) | |
|---|---|---|---|---|
| | | | Positive N (%) | Negative N (%) |
| Sex | Male | 2006(45.2) | 494(24.6) | 1512(75.4) |
| | Female | 2435(54.8) | 578(23.7) | 1857(76.3) |
| Age | ≤ One year | 307(6.9) | 64(20.8) | 243(79.2) |
| | 2–10 years | 702(15.8) | 108(15.4) | 594(84.6) |
| | 11–20 years | 555(12.5) | 109(19.6) | 446(80.4) |
| | 21–30 years | 1205(27.1) | 268(22.2) | 937(77.8) |
| | 31–40 years | 554(12.5) | 123(22.2) | 431(77.8) |
| | 41–50 years | 384(8.6) | 97(25.3) | 287(74.7) |
| | 51–60 years | 285(6.4) | 96(33.7) | 189(66.3) |
| | 61–70 years | 228(5.1) | 108(47.4) | 120(52.6) |
| | 71 and above | 221(5) | 99(44.8) | 122(55.2) |
| | Total | 4441(100) | 1072(24.1) | 3369(75.9) |

Key: Frequency (%) column is calculated from the total sample size (4441); UTI rate of each category of variables is calculated row wise.

**Table 2. The proportion of uropathogenic isolates at the university of Gondar comprehensive specialized hospital, 2010–2019.**

| Species of isolates | Frequency | Percentage (from total UTI patients; 1072) |
|---|---|---|
| *E. coli* | 404 | 37.7 |
| *K. pneumoniae* | 122 | 11.4 |
| CONS | 46 | 4.3 |
| *S.aureus* | 98 | 9.14 |
| *Prouteus* Spp. | 17 | 1.6 |
| *Citrobacter* Spp | 76 | 7.1 |
| *Salmonella* Spp | 6 | 0.6 |
| GNR | 111 | 10.34 |
| *Enterobacter* Spp | 37 | 3.45 |
| *Streptococcus* Spp | 26 | 2.42 |
| *Klebsiella* Spp | 36 | 3.35 |
| *Pseudomonas* Spp | 6 | 0.55 |
| *Providencia* Spp | 5 | 0.46 |
| *K.ozaenae* | 45 | 4.2 |
| *Shigella* Spp | 8 | 0.74 |
| *Enterococcus* Spp | 23 | 2.14 |
| *Serratia* Spp | 4 | 0.37 |
| *M.morgani* | 2 | 0.18 |

Key: CONS- Coagulase Negative Staphylococci, GNR-Gram negative rods, Spp- Species.

isolates showed 88.9%, 83.6%, 76.5%, and 74% resistance to ampicillin, tetracycline, trimethoprim-sulfamethoxazole, and amoxicillin-clavulanate, respectively. All Gram-negative isolates had a high ampicillin resistance ranging from 89–100% (**Table 4**).

## The yearly basis of resistance pattern of the classes of antimicrobial agents

The resistance rate of isolated bacteria was seen to be inconsistent to the majority of the tested antimicrobial classes. However, the resistance to cephalosporins has been rising, particularly from 2014 to 2019 (Table 5).

**Table 3. The proportion of resistant Gram-positive isolates among UTI patients at the University of Gondar comprehensive specialized hospital, 2010–2019.**

| Antibiotics | CONs N (%) | *S. aureus* | *Streptococcus* spp N (%) | *Enterococcus* spp N (%) | Row Total N (%) |
|---|---|---|---|---|---|
| AMP | ND | ND | 4/8(50) | 7/7(100) | 11/15(73.3) |
| PEN | 16/19(84.2) | 22/32(68.8) | 7/12(58.3) | 3/4(75) | 48/67(71.6) |
| CIP | 13/20(65) | 36/57(63.2) | 7/15(46.7) | 12/20(60) | 68/112(60.7) |
| GEN | 9/24(37.5) | 13/30(43.3) | 7/11(63.6) | 4/7(57.1) | 33/72(45.8) |
| NIT | 2/5(40) | 3/21(14.3) | 1/10(10) | 6/12(50) | 12/48(25) |
| NOR | 11/15(73.3) | 24/39(61.5) | 4/6(66.7) | 5/7(71.4) | 44/67(65.7) |
| TET | 22/26(84.6) | 48/56(85.7) | 9/12(75) | 5/5(100) | 84/99(84.8) |
| CAF | 9/18(50) | 11/29(37.9) | 0/9(0) | 1/6(16.7) | 21/62(33.9) |
| FOX | 1/1(100) | 3/10(30) | 1/4(25) | 2/2(100) | 7/17(41.2) |
| SXT | 17/19(89.5) | 33/41(80.5) | 12/13(92.3) | 9/12(75) | 71/85 (83.5) |
| CRO | 9/25(36) | 11/35(31.4) | 6/11(54.5) | 2/4(50) | 28/75(37.3) |
| VAN | ND | ND | 1/8(12.5) | 2/6(33.3) | 3/14(21.4) |

Key: AMP-ampicillin, PEN- penicillin, AMC-amoxicillin-clavulanic acid, CIP- ciprofloxacin, GEN-gentamycin, NIT- nitrofurantoin, NOR- norfloxacin, TET-tetracycline, TOB- tobramycin, FOX- cefoxitin, CRO- ceftriaxone, VAN-vancomycin, NA- nalidixic acid, OXA-oxacillin, ND-not done.

Table 4. The proportion of resistant Gram-negative isolates among UTI patients at the University of Gondar comprehensive specialized hospital, 2010–2019.

| Antibiotics | E. coli N (%) | K.pneu moniae N (%) | Proteus spp N (%) | Citrobacter spp N (%) | Salmon ella spp N(%) | NLFGNR N (%) | Enterob acter spp N (%) | Klebsiella spp N (%) | LFGNR N (%) | Pseudo monas spp N (%) | Provid encia spp N (%) | K.ozaena e N (%) | Shigell a spp N (%) | Serratia spp N (%) | Row Total N (%) |
|---|---|---|---|---|---|---|---|---|---|---|---|---|---|---|---|
| AMP | 219/246 (88.9) | 64/65 (98.5) | 12/13 (92) | 48/49 (98) | 3/4 (75) | 32/35 (91.4) | 19/20 (95.0) | 27/27 (100) | 26/29 (89.6) | ND | 2/2(100) | 21/22 (95.5) | 4/4 (100) | 2/2(100) | 479/518 (92.5) |
| AMC | 128/173 (74) | 44/51 (86.3) | 5/9 (55.6) | 31/32 (96.9) | 3/3 (100) | 20/27 (74) | 9/10 (90.0) | 15/16(93.8) | 16/21 (76.2) | ND | 2/2(100) | 17/19 (89.5) | 2/2 (100) | 1/1(100) | 293/366 (80.1) |
| CIP | 155/279 (55.6) | 30/78 (38.5) | 8/15 (53.3) | 29/57 (50.9) | 1/4 (25) | 20/44 (45.5) | 13/20(65) | 15/27(55.6) | 15/32 (46.9) | 1/6(16.7) | 2/5(40) | 15/26 (57.7) | 5/7(71.4) | 1/2(50) | 309/602 (51.3) |
| NOR | 122/179 (61.9) | 19/48 (40) | 4/7 (57.1) | 14/34 (41.2) | 1/2 (50) | 18/33 (54.5) | 7/13(53.8) | 7/17(41.2) | 17/26 (65.4) | 1/4(25) | 2/5(40) | 14/20(70) | 3/5(60) | 1/2(50) | 230/395 (58.2) |
| NA | 45/69 (65.2) | 18/29 (66.7) | 1/1 (100) | 9/14 (64.3) | ND | 7/12 (58.3) | 3/8(37.5) | 8/11(72.7) | 10/13 (76.9) | ND | ND | 7/11(63.6) | 1/1(100) | 1/1(100) | 110/170 (64.7) |
| TOB | 33/72 (45.8) | 18/29 (62.1) | 1/5 (20) | 7/14 (50) | ND | 6/9 (66.7) | 3/4(75) | 1/4(25) | 2/7(28.6) | 1/1(100) | ND | 2/9(22.2) | 1/3(33.3) | 0/1 | 75/158 (47.5) |
| GEN | 103/240 (42.9) | 35/58 (58.6) | 7/13 (53.8) | 20/36 (55.6) | 2/3(66.7) | 19/30 (63.3) | 14/17(82.4) | 8/15(53.3) | 16/29 (55.2) | 1/4(25) | 2/4(50) | 15/22 (68.2) | 3/5(60) | ND | 245/476 (51.5) |
| AMK | 6/25 (24) | 4/11 (36.4) | ND | 0/11 | ND | 2/3 (66.7) | 0/5 | 0/2 | 0/3 | 0/1 | ND | 1/4(25) | ND | ND | 13/65 (20) |
| KAN | 7/13 (53.8) | 5/7 (71.4) | ND | 1/1 (100) | 1/1(100) | 1/1 (100) | 1/1(100) | ND | 1/2(50) | ND | ND | 1/2(50) | 1/4(25) | ND | 19/32 (59.4) |
| NIT | 21/133 (15.8) | 13/41 (31.7) | 4/4 (100) | 11/21 (52.4) | ND | 12/24 (50) | 2/7(28.6) | 6/7(85.7) | 5/17(29.4) | 3/3(100) | ND | 0/11 | 0/1 | 1/1 (100) | 78/270 (28.9) |
| SXT | 186/243 (76.5) | 55/66 (83.3) | 8/10 (80) | 29/37 (78.4) | 2/3(66.7) | 33/38 (86.8) | 15/17(88.2) | 16/20(80) | 15/21 (71.4) | 3/5(60) | 3/3(100) | 24/29 (82.8) | 3/7(42.9) | 1/3 (33.3) | 393/502 (78.3) |
| TET | 163/195 (83.6) | 33/46 (71.7) | 10/11 (91) | 31/40 (77.5) | 0/1 | 24/29 (82.8) | 14/18(77.8) | 16/18(88.9) | 19/24 (79.2) | 2/3(66.7) | 4/5(80) | 12/18 (66.7) | 2/5(40) | 2/2 (100) | 332/415 (79.3) |
| CAF | 43/163 (26.4) | 22/38 (57.9) | 8/10 (80) | 17/32 (53.1) | 1/2(50) | 12/20 (60) | 9/15(60) | 5/13(38.5) | 9/18(50) | 2/2(100) | 1/4(25) | 9/15(60) | 2/5(40) | 1/1 (100) | 141/338 (41.7) |
| CRX | 26/38 (68.4) | 15/17 (88.2) | 1/3 (33.3) | 5/5 (100) | 1/1(100) | 6/6 (100) | 3/3(100) | 2/2 (100) | 4/6(66.7) | 1/1(100) | ND | 8/9(88.9) | 1/1(100) | ND | 73/92 (79.3) |
| CRO | 93/205 (45.4) | 33/54 (61.1) | 5/14 (35.7) | 27/41 (65.9) | 2/3(66.7) | 18/27 (66.7) | 9/11(81.8) | 17/24(71) | 14/22 (63.6) | 1/2(50) | 2/4(50) | 15/17 (88.2) | 3/6(50) | 1/2(50) | 240/432 (55.6) |
| FOX | 6/18 (33.3) | ¾ (75) | ND | 3/5 (60) | ND | 1/1(100) | 1/2(50) | 2/3(66.7) | 2/3(66.7) | 1/1(100) | ND | 1/2(50) | ND | ND | 20/39 (51.3) |
| CTX | 4/10 (40) | 4/4 (100) | ND | 0/4 | 1/1(100) | 0/1(0) | 2/3(66.7) | 0/1 | 2/2(100) | ND | ND | ND | ND | ND | 13/26 (50) |
| CAZ | 14/25 (56) | 6/10 (60) | 0/2 | 1/4 (25) | ND | 3/3 (100) | 2/3(66.7) | 1/3(33.3) | 3/6(50) | ND | ND | 5/5(100) | 0/1 | ND | 35/62 (56.5) |
| MER | 9/34 (26.5) | 3/17 (17.6) | 0/2 | 2/7 (28.6) | ND | 4/9 (44.4) | 4/5(80) | 0/2 | 0/3 | ND | ND | 2/9(22.2) | 0/3 | ND | 24/91 (26.4) |

Key: AMP-ampicillin, AMC-amoxicillin-clavulanic acid, CIP- ciprofloxacin, GEN-gentamycin, NIT- nitrofurantoin, NOR- norfloxacin, AMK-amikacin, KAN-Kanamycin, TET- tetracycline, TOB-tobramycin, FOX- cefoxitin, CRO- ceftriaxone, NA- nalidixic acid, CRX-cefuroxime, CTX-cefotaxime, CAZ-ceftazidime, MER- meropenem, SXT-Trimethoprim-Sulfamethoxazole, CAF-chloramphenicol, LFGNR-Lactose fermenting Gram negative rods, NLFGNR- None lactose fermenting Gram negative rods, ND-not done, N- Number of organisms of a specified species tested to each antimicrobial agent. Numerators represent number of resistant bacteria to each antimicrobial while the denominators are the number of bacteria tested against each antimicrobial agent (both resistant and susceptible).

**Table 5. Yearly basis of antimicrobial resistance to classes of antimicrobial agents, 2010–2019.**

| Class | Number N (%) of isolates tested to each class of antibiotics | | | | | | | | | | |
|---|---|---|---|---|---|---|---|---|---|---|---|
| | 2010 | 2011 | 2012 | 2013 | 2014 | 2015 | 2016 | 2017 | 2018 | 2019 | Total |
| Penicillin | 133/ 144 (92.4) | 98/114 (86) | 148/ 175 (84.6) | 109/ 119 (91.6) | 81/89 (91) | 71/79 (90) | 35/43 (81.4) | 42/48 (87.5) | 75/97 (77.3) | 49/58 (84.5) | 841/ 966 (87.1) |
| Fluoroquinolone | 97/177 (54.8) | 67/125 (53.6) | 104/ 174 (59.8) | 92/165 (55.8) | 78/ 126 (61.9) | 42/87 (48.3) | 29/65 (44.6) | 83/ 133 (62.4) | 93/ 160 (58.1) | 80/ 129 (62) | 765/ 1341 (57) |
| Aminoglycoside | 54/100 (54) | 29/58 (50) | 10/23 (43.5) | 39/61 (63.9) | 33/68 (48.5) | 30/63 (47.6) | 14/24 (58.3) | 26/52 (50) | 93/ 222 (41.9) | 57/ 122 (46.7) | 385/ 793 (48.5) |
| Cephalosporin | 44/56 (78.6) | 32/43 (74.4) | 27/56 (48.2) | 54/94 (57.4) | 34/76 (44.7) | 30/56 (53.6) | 14/25 (56) | 11/19 (57.9) | 90/ 127 (70.9) | 80/ 113 (70.8) | 416/ 665 (62.6) |
| Folate pathway inhibitor (SXT) | 80/95 (84.2) | 51/69 (73.9) | 15/22 (68.2) | 26/37 (70.3) | 35/42 (83.3) | 37/53 (69.8) | 30/36 (83.3) | 60/69 (87) | 62/79 (78.5) | 68/85 (80) | 464/ 587 (79) |
| NIT | ND | ND | ND | ND | 9/29 (31) | 9/49 (18.4) | 4/28 (14.3) | 8/40 (20) | 37/ 109 (33.9) | 23/63 (26.7) | 90/318 (28.3) |
| TET | 79/102 (77.5) | 61/77 (79.2) | 67/81 (82.7) | 76/91 (83.5) | 39/49 (79.6) | 49/56 (87.5) | 21/29 (72.4) | 5/7 (71.4) | 13/15 (86.7) | 6/7 (85.7) | 416/ 514 (80.9) |
| CAF | 41/97 (42.3) | 26/66 (39.4) | 7/23 (30.4) | 27/43 (62.8) | 25/78 (32.1) | 12/29 (41.4) | 3/15 (20) | 12/31 (38.7) | 5/10 (50) | 4/8 (50) | 162/ 400 (40.5) |
| Carbapenem | ND | ND | ND | ND | ND | ND | ND | ND | 13/53 (20.8) | 11/38 (28.9) | 24/91 (26.4) |
| Glycopeptide (VAN) | ND | ND | ND | ND | ND | ND | ND | ND | 2/9 (22.2) | 1/5 (20) | 3/14 (21.4) |

ND- Antimicrobial susceptibility test was not done, VAN- vancomycin.

## Antimicrobial resistance (AMR) trend and multidrug resistance (MDR) rate of isolates

In the last ten consecutive years (2010–2019), the antimicrobial resistance trend of uropathogens ranges from 50 to 66.5%. The resistance rate was highest (66.5%) in 2012, and the lowest (50.2%) observed resistance was in 2016. From 2012 to 2016, there was a reduction in antibiotic resistance rates. However, there was a slight increment in the resistance rate from 2016 to 2019 (Fig 1).

Multidrug-resistant (MDR) isolates were 473 (44.1%) of the total 1072 uropathogens. *E. coli* 199 (42.1%) and *K. pneumonia* 51(10.8%) were the predominant MDR uropathogens, which together account for more than half of the total MDR isolates. Among Gram-positive uropathogens, *S. aureus* 31(6.6%) was the leading MDR isolate, placed 4[th] among all MDR isolates (Fig 2).

## Discussion

Globally, human health is in danger from antimicrobial-resistant infection. To strengthen knowledge of AMR through surveillance and research World Health Organization (WHO) opened a program called the Global Antimicrobial Resistance Surveillance System (GLASS). *Escherichia coli*, *K. pneumoniae*, and *S. aureus*, which are the leading etiologic agents of UTI

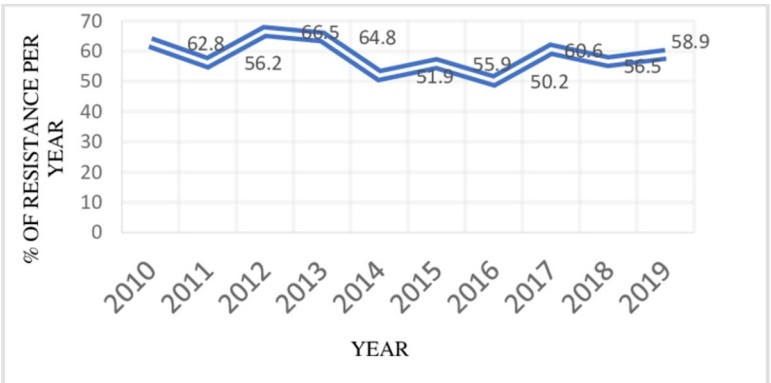

**Fig 1. The overall trend of antimicrobial resistance of isolates to the tested antibiotics among UTI patients at the University of Gondar comprehensive specialized Hospital, 2010–2019.**

and common resistant bacteria, are among the GLASS targets [19]. Epidemiological studies with time in different geographic regions are the first and vital steps for selecting effective antimicrobial agents for treatment, preventive and control actions [20]. The overall prevalence of UTI in this study was 1072(24.1%) [95% CI, (22.9–25.4)], with a proportional infection rate in males (24.6%) and females (23.7%). This finding is consistent with results from studies conducted in Dessie (22.7%) [21] and Addis Ababa, Ethiopia (23.32%) [22], as well as other parts of the world such as India (22.8%) [23] and central Europe (26.9%) [24]. The age-wise distribution of the infection reveals 47.4% UTI in the elderly population above 60 years. In contrast, according to a study done in Addis Ababa the most affected age group of participants was the age group 21–30 years [22]. Similar to our study, an increased infection rate in the elderly participants has been reported elsewhere [16]. The elderly population might be at risk of acquiring UTI due to age-related weakened immunity, change in vaginal hormonal secretion, or other comorbidities [25]. On the other hand, the rate of UTI in our study was lower than studies

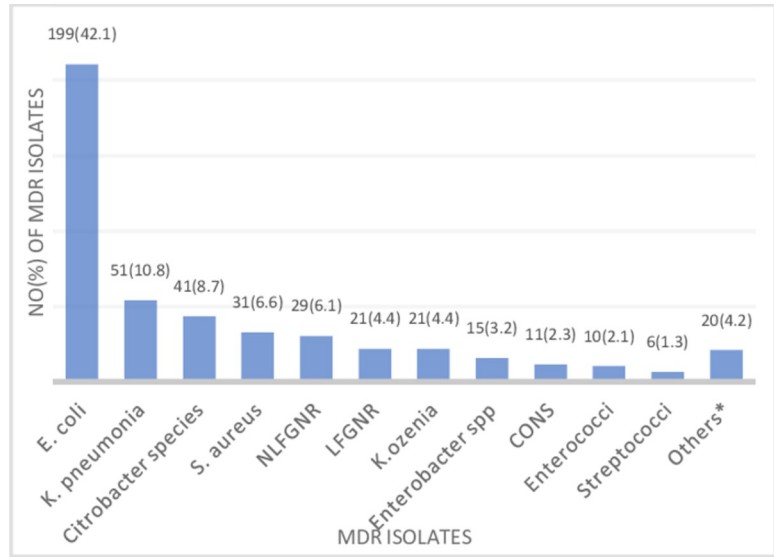

**Fig 2. Frequency of MDR species isolated in 10 years retrospective study (473/1072 = 44.1%).** Key: Proteus 7(1.5%), shigella 5(1.1%), Providencia 3(0.6%), Pseudomonas 2(0.4%), Salmonella, Serratia, *M. morganii* 1(0.2%) each.

conducted in Bahir Dar, Ethiopia (30.5%) [26], India (38.84%) [27], India ((28.2%) [28], Sudan (91%) [29], but higher than the prevalence studies conducted in Iran (15%) [20] and India (15.9%) [30]. Geographic and population differences, study design, or the laboratory method used among studies may explain the difference in the prevalence rate.

Regarding the proportion of bacterial isolates in this study, *E. coli* 404 (37.7%) was the leading isolate among uropathogens, followed by *K. pneumoniae* (11.4%) and *S. aureus* (9.14%). Similar findings have been reported in different geographic regions; Gondar [16], Bahir Dar [26] and Hawassa [31], and Addis Ababa, Ethiopia [27], and India [28].

The resistance rate of Gram-positive isolates was high to agents such as tetracycline (84.8%), trimethoprim-sulfamethoxazole (83.5%), fluoroquinolones (60.7–65.7%), and penicillin (71.6%). This figure is much higher than the resistance rate (34.6%) reported in Iran [20]. Moreover, more than 75% of Gram-negative isolates in this study, were resistant to ampicillin (92.3%), amoxicillin-clavulanate (80.1%), tetracycline (79.3%), cefuroxime (79.2%), and Trimethoprim-sulfamethoxazole (78.3%). *Escherichia coli* isolates were highly resistant to ampicillin (88.9%), tetracycline (83.6%), trimethoprim-sulfamethoxazole (76.5%), and amoxicillin-clavulanate (74%). The resistance rate continues to grow even for those antimicrobial agents including cefuroxime (68.4%) which are limited to selected tertiary hospitals. Comparable resistance rate among isolates were reported in Sudan; ampicillin (94%), amoxicillin-clavulanate (90%), tetracycline (76%), norfloxacin (74%), trimethoprim-sulfamethoxazole (88%) and ceftriaxone (68%) [31]. In this study, more than half of the Gram negative isolates were resistant to fluoroquinolones which outnumbers a report in the USA (24.3%-25.8%) [32]. Moreover, nitrofurantoin (35%), ciprofloxacin (28.8%), and ceftriaxone (25.9%) were better agents for uropathogens in another study [33].

In this study, 26.4% and 20% of Gram-negative isolates were resistant to meropenem and amikacin respectively. It is worrisome because these agents were considered the most effective agents in treating UTIs [32]. Furthermore, the rate of resistance of uropathogens to amikacin (20%), ciprofloxacin (51.3%), and cefuroxime (79.3%) in this study were lower than the resistance rate reported in Turkey [34]. Different reports showed that uropathogens are highly resistant to ampicillin, amoxicillin-clavulanate, and trimethoprim-sulfamethoxazole [33, 34]. The recommendations from national guidelines for antimicrobial use in different countries could have resulted in varying resistance among countries [35]. The resistance trend of uropathogenic isolates over ten consecutive years (January 2010–2019) ranges from 50.2% in 2016 to 66.5% in 2012. The resistance rate of isolated bacteria to antimicrobial classes was inconsistent but the resistance to cephalosporins has shown increasing pattern particularly from 2014 to 2019. Absence of uniform supply and hence, irregular use of these antibiotics in the laboratory can be mentioned for such inconsistent trend of resistance. The rising resistance to cephalosporins may be due to the rising preference and clinical use of this agent.

This study showed an inconsistent trend of the overall resistance rate with a reducing rate between 2012 and 2016, while a slight increment from 2016 to 2019. In general, the overall resistance rate surpasses 50% over the study period (**Fig 1**). Moreover, there were vancomycin-resistant *Enterococcus* species (33.3%) in this study, which was a higher prevalence than a report (14.8%) by Melese et al. in Ethiopia [36] but lower than 54% resistance reported in Sudan [29]. In addition, 6(50%) nitrofurantoin resistant *Enterococcus* species were isolated in this study which was higher than (9.8%) vancomycin and (0–40%) nitrofurantoin resistant *Enterococcus* species reported in England [37]. Methicillin-resistant *S. aureus* (MRSA) is an emerging threat evolving rapidly, and 30% of MRSA strains were isolated in this study which needs an urgent response [38].

Multidrug resistance is a concern of the medical community because there is a run out of effective antimicrobial agents to relieve the suffering of patients and save lives [39]. We found

a total of 473(44.1%) MDR UTI which concords to a result reported in Tunisia (45.1%) [40]. However, our finding was higher than 25% in Portugal [41] and 36.5% in Germany [42]. The result of this study otherwise, was lower compared to results from Hawassa, southern Ethiopia (80.3%) [31], in Serra Leone (85.7%) [43], Saudi Arabia (80%) [1] and Serbia (53.8%) [44]. The Geographic variation, limited activity towards implementation of antimicrobial stewardship program, and a different definition of MDR might contribute to the observed differences. The species *E. coli* (42.1%), *K. pneumonia* (10.8%), *Citrobacter* species (8.7%), and *S. aureus* (6.6%) were the most common MDR isolates in this study. This result was lower than the 79.3% MDR in southern Ethiopia among HIV patients [45], who frequently take antibiotics and are at higher risk of MDR infections. In general, MDR, Carbapenem resistance, MRSA, and vancomycin resistance have been observed in this study. Hence, healthcare professionals and other stakeholders need to be curious about the supply and control of antimicrobial agents as we are on the verge of loss of effective agents [20]. Considering the existing resistance in the hospital from our result, healthcare providers in the hospital would selectively use the effective antibiotics. The global community would understand the burden of resistance in the area and design comprehensive policies by aggregating with reports from diverse geographic settings. Limitation, as the study was a retrospective study, we did not analyze factors contributing to resistance. The resistance trend of each species was not determined because of inconsistent use of antimicrobials to each species. In addition, it is a single-center hospital-based study among symptomatic patients. We were also not able to differentiate between inpatients and out patients and compare the rate of UTI. As the laboratory did not have any molecular based detection, this study was limited to conventional culture-based results.

## Conclusions

The overall of prevalence of UTI was high and elderly patients were most affected. *Escherichia coli* was the most common bacterial uropathogen, followed by *K. pneumonia* and *S. aureus*. Multidrug resistance in this study is high, which alarms a need for problem resolution using routine diagnosis and antimicrobial susceptibility testing rather than empirical treatment. A continuous revision of UTI treatment guideline to replace ineffective agents with potent alternatives should be done. A prospective multicenter study including asymptomatic population is indispensable to know status of resistance.

## Acknowledgments

We would like to acknowledge the Microbiology laboratory and the hospital management of UoGCSRH for their genuine cooperation to ensure this research work.

## Author Contributions

**Conceptualization:** Blen Desalegn, Mihret Aynalem, Dureti Diriba, Michael Getie.

**Data curation:** Mihret Aynalem, Dureti Diriba, Beimnet Afework.

**Formal analysis:** Desie Kasew, Sosina Tila, Beimnet Afework.

**Funding acquisition:** Sosina Tila, Dureti Diriba, Beimnet Afework.

**Investigation:** Desie Kasew, Blen Desalegn, Mihret Aynalem, Sosina Tila, Dureti Diriba, Beimnet Afework, Sirak Biset, Habtamu Wondifraw Baynes.

**Methodology:** Desie Kasew.

**Project administration:** Desie Kasew, Sosina Tila, Beimnet Afework, Michael Getie.

**Resources:** Blen Desalegn, Beimnet Afework, Habtamu Wondifraw Baynes.

**Software:** Mihret Aynalem, Sirak Biset.

**Supervision:** Michael Getie, Sirak Biset, Habtamu Wondifraw Baynes.

**Validation:** Desie Kasew, Dureti Diriba, Michael Getie, Sirak Biset, Habtamu Wondifraw Baynes.

**Visualization:** Desie Kasew, Michael Getie, Sirak Biset, Habtamu Wondifraw Baynes.

**Writing – original draft:** Desie Kasew, Blen Desalegn, Sosina Tila.

**Writing – review & editing:** Michael Getie, Sirak Biset, Habtamu Wondifraw Baynes.

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
