## [Decision Letter · Decision Letter 0]

13 Aug 2021

PONE-D-21-22966

Antimicrobial Resistance Trend of Bacterial Uropathogens at the University of Gondar Comprehensive Specialized Hospital, Northwest Ethiopia: A 10 years retrospective study

PLOS ONE

Dear Dr. Kasew,

Thank you for submitting your manuscript to PLOS ONE. After careful consideration, we feel that it has merit but does not fully meet PLOS ONE’s publication criteria as it currently stands. Therefore, we invite you to submit a revised version of the manuscript that addresses the points raised during the review process.

Please read carefully the reviewers comments and suggestions specially those from reviewer 2 and resubmit the revised version as early as your convenience.

We look forward to receiving your revised manuscript.

Kind regards,

Monica Cartelle Gestal, PhD

Academic Editor

PLOS ONE

Journal Requirements:

"No: The funders had no role in study design, data collection and analysis, decision to publish, or preparation of the manuscript."

5. Please note that in order to use the direct billing option the corresponding author must be affiliated with the chosen institute. Please either amend your manuscript to change the affiliation or corresponding author, or email us at plosone@plos.org with a request to remove this option.

6. Please amend your list of authors on the manuscript to ensure that each author is linked to an affiliation. Authors’ affiliations should reflect the institution where the work was done (if authors moved subsequently, you can also list the new affiliation stating “current affiliation:….” as necessary)

Reviewers' comments:

Reviewer's Responses to Questions

**Comments to the Author**

1. Is the manuscript technically sound, and do the data support the conclusions?

Reviewer #1: No

Reviewer #2: Partly

2. Has the statistical analysis been performed appropriately and rigorously? 

Reviewer #1: No

Reviewer #2: N/A

3. Have the authors made all data underlying the findings in their manuscript fully available?

Reviewer #1: No

Reviewer #2: Yes

4. Is the manuscript presented in an intelligible fashion and written in standard English?

Reviewer #1: No

Reviewer #2: Yes

5. Review Comments to the Author

Reviewer #1: dear authors

there are several lacunae in the methodology part with respect to inclusion and exclusion criteriae. more than one loose statements have been made without citations. quite a lot of errors seen in the discussion section too. kindly provide the details of analysis made for making out the the trends in antimicrobial resistance over years. kindly go through the comments and make the necessory revisions

Reviewer #2: General Comment:

Interesting topic about AMR, observing 10 years laboratory-data to establish the local guidance. Enormous number of isolate collecting in single hospital as study site.

Specific Comment:

In method section, sub-section Study area (Line 92), it is stated “We have collected ten years (2010-2019) retrospective data from the microbiology laboratory logbook”. This should describe more detailed e.g mention how does the author confirm the adequate data and represent for each year period? Moreover, the author should also mention the consistency of practical work in the laboratory for ten years, i.e. method of susceptibility testing, antimicrobial disk used. I think this is very important point as the strength of the manuscript.

In method section, sub-section population (Line 94-95), it is stated “all symptomatic patients who visited the hospital from 2010-2019 and had recorded urine culture results were the study population”. Please describe how does the author perform the screening of the patient!

This is important information to be added in the method section. Moreover, description about elaboration data with microbiology aspect is needed, as appear in Line 129-130: “The data were collected by investigators with quality and completeness checks throughout the collection period, at the end of data collection, and after entry to SPSS for statistical analysis.” Also, need to define the definition for trend of AMR in this study!

In method section, the author should mention how the percentage calculation for resistance trend among antibiotic tested. What is denominator of the calculation? This information is still missing from the method section.

In the inclusion criterion, I think it seems like “negative” statement or opposite from inclusion, please clarify!

In Line 110 – 121: Does it show microbiology procedure? Please write more clearly and elaborate with whole process of the study.

Result:

In line 134, the author mentioned about “UTI suspected patient”; but in population section, it was described “all symptomatic patients” (Line 94-95), which means not suspected patient. Please clarify for this sentence! If the author started with suspected patient, then provide the number of patient being suspected and mentioned how many of this become symptomatic patient with culture-proven UTI.

In table 2: the percentage is confusing and need to defined properly in method section (see the method comment). There is number of 4441, which indicate the number of total specimen, and number of 1072, which indicate the isolate included based on study criteria. Please explain for this two-different information!

Lot of mistype for species name i.e Prouteus, Klebsella. Author need to check all the typing for species name. Also, consistency for using italic formatting.

In table 3 and 4: what does the number in bracket indicate? Please provide clear calculation for this table. What is the denominator for the calculation?

I think the figure 1 and 2 is main finding of this study and should be attached in the body of manuscript, instead as supplementary figure. the result should more address the trend of bacterial and antimicrobial from 10 years observation, instead of mentioned single species with resistance profile.

Discussion:

I don’t see the discussion related with trend as the main title of the manuscript. Please correlate the finding with the term of Trend as mentioned with title of this study.

6. PLOS authors have the option to publish the peer review history of their article (what does this mean?). If published, this will include your full peer review and any attached files.

Reviewer #1: No

Reviewer #2: No

---

## [Author Response · Author response to Decision Letter 0]

17 Sep 2021

Response to reviewer(s) 

A. To PLOS ONE RVIEW

We are pleased that you, the PLOS ONE editorial team, offered us an invitation to revise our manuscript for the PLOS ONE journal. Also, we appreciate you for assigning such talented and qualified reviewers to our manuscript. We believed that their valuable comments helped us to improve our work.

1. Line 60- Pyelonephritis and kidney infection are almost same. 

Response: Yeah, these terms are referring to infections of the upper urinary tract. We have revised and corrected it.

2. Line 61.62- suprapubic tenderness is a sign. 

Response: “Suprapubic tender ness” has been included in a list of signs in many articles including the cited article. But now we have amended as sign and symptoms.

3. 62.63- could be corrected as bacterial causes as UTI can be due to fungal species like candida

Response: Thanks for your suggestion, and now we have amended it. 

.

4. 68- Adherence to urethral mucosa? is it unique for female sex alone?

Response: Thank you for raising this issue. It is not unique for female sex alone, therefore, we have amended the sentence

5. 69- Kindly clarify the usage of the term “directly associated”. Doesn’t sound right.

Response: Thanks! We have revised and amended. As UTIs are among the most frequently treated infections, especially among women, it is obvious that exposure or usage of antibiotics will be increased, which intern hastens the emergence of antibiotic resistance. Hence these infections and drug resistance can have direct association. 

6. 74- You mean Ethiopia alone? 

Response: No, we don’t mean that Ethiopia is the only one affected by these factors. The mentioned problems are also reported in many countries. These practices have been widely perpetuated in Ethiopia because of loose control system and wide availability of drugs without prescription in Ethiopia. There is a minimal attention to prevention of antimicrobial resistance in the country and awareness about the misuse or/and overuse of antibiotics in the community. 

7. 78- Suggest usage of appropriate empirical treatment rather than appropriate treatment alone as most UTIs are treated empirically based on prevalent antimicrobial pattern.

Response: Thank you for your suggestion, and now we have amended it. 

8. 81.82.83-sentence can be reframed to include trend over 10 years between 2009 and 2019. Response: Amended as suggested but it is from 2010 to 2019, not 2009 to 2019.

9. 86- study period should include the period from which the study sample is selected rather than the period at which the data is collected.

Response: The period is included as per the comment and the sentence is written as “A Hospital-based retrospective cross-sectional study was conducted by retrieving laboratory record of ten years (2010-2019) data from January 1st to February 28th, 2020 at the UoGCSH, Gondar, Ethiopia”

10. 92- Was it a manual retrieval or retrieval from computerized data?

Response: We have retrieved the data manually from the Laboratory paper-based record book

11. 93, 94- did the study included patients of all ages from birth to the maximum age recorded? If so any difference in the criteria for interpreting the results was applied between pediatric UTI and adult UTI? Did the study include outpatients or inpatients or both?

Response: Yes, this study included UTI patients irrespective of age group as per the inclusion criteria (having a complete record of the variables mentioned in the inclusion criteria). There were not any different criteria for interpreting UTI between pediatric and adults or any other age group. The microbiology laboratory at the UoGCSH uses growth of 105 CFU/ml as a criterion for defining significant bacteriuria. We have used a record-based data and not able to identify between outpatients and inpatients. We could not determine the number of outpatients and inpatients as the origin of specimen was not recorded in the laboratory result registration book. We know that the UoGCSH laboratory receives urine for culture from inpatient and outpatients, therefore, we believe that both inpatients and outpatients were included in this study.

12. 96- What was the total number of samples recorded in the log book and how many were excluded based on your exclusion criteria. how many were excluded because of lack of information like age and sex in spite of having positive culture reports, which can influence the analysis? 

Response: We have collected and analyzed records, which have full data of our target variables but the total number of the recorded urine culture results were 4209, of which 4088(included for analysis in this study) and 121 (rejected from this study because of missing of one or more of our inclusion criteria). Herewith we have mentioned the number and reason for exclusion in a table below.

Number of samples rejected N=121 Reason for rejection

7 Age was not recorded or missed

18 Sex was not recorded or missed

83 Age and sex were not recorded or missed

11 Culture result was not recorded

2 Isolated organism was not recorded

13. 111- “organisms” would be better term rather than” colonies” with significant 

Response: Thank you for your suggestion. We have changed it to “organisms”.

14. 125- Which version of CLSI guidelines? As the guidelines get revised periodically, kindly be more specific. 

Response: Yes, it is known that CLSI is revised in a year basis, and the laboratory uses yearly uprated versions of the CLSI. So, a single year CLSI guideline has not been used. 

15. 127. “The quality and completeness of patients’ urine culture records, as well as antimicrobial susceptibility results, were checked”. Can you clarify the statement as it is repeated again in the next sentence?

Response: Thank you for pointing this out. We have corrected it.

16. 135 UTI suspected patients? According to the exclusion criteria patients without antimicrobial susceptibility testing were excluded from the study. Then how come they are included in the results? Kindly clarify.

Response: We used “UTI suspected patients” to express those patients with urinary tract complain and for whom urine culture test was requested.

Patients without antimicrobial susceptibility testing result were excluded from the study which mean patient with significant bacteriuria but no AST recorded. This means for patients with significant bacteriuria, although records of age, sex, and significant bacteriuria were available, they were excluded if records of their antimicrobial susceptibility test results were not available, if result was “significant bacteriuria”, AST must also be recorded to include into the study. This does not mean that the isolate must be tested to all the mentioned antimicrobial agents. The isolated organisms were tested to different agents depending on their availability in the laboratory. Records without significant bacteriuria were also included if all variables age, sex and culture result (no significant bacteriuria) were recorded. 

17. 152,153- There is no information about antibiotic susceptibility in table 2 while it has been cited here. Highly irrelevant.

Response: Thank you! We have modified it. 

For understanding we have mentioned that “only two Morganella morganii (M. morganii), both resistant to ampicillin and amoxicillin-clavulanate, and either of which being resistant to gentamycin, trimethoprim-sulfamethoxazole, tetracycline, ciprofloxacin, tobramycin, and nalidixic acid were recorded (Table 2)”

Here, the antimicrobial susceptibility test was not written because only 2 Morganella morganii isolates were isolated and tested and their susceptibility results for the tested antibiotics was as mentioned in the text. Their susceptibility was omitted from the table 4 to minimize crowding. The table 2 was cited for the species of isolated organisms, not to the antimicrobial susceptibility testing. But now we have canceled it to avoid confusion.

18. 156 table took looks very irrelevant what is the meaning of calculating the percentage with the overall urinary samples when it is already calculated for positive cultures alone? 

Response: We have used Table -2 to indicate the proportion of each bacterial species from the total isolates. However, as it is suggested, the proportion from the total number of urine sample (column 3) could have been omitted and now corrected.

19. 185 table could have been better with the explanation for numbers and percentages in the heading row (does it mean the percentage positive out of total numbers tested?)

Response: Thank you for your suggestion. We have reported the resistant isolates using frequencies and percentages. Each percentage were calculated by dividing the number of the same bacteria (e.g., E. coli) reported as resistant to a particular antibiotic agent (e.g., Ceftriaxone) from the total number of that bacteria (resistant or sensitive) tested against the same agent. 

20. 192- kindly explain the methodology by which this trend is obtained in the methodology section itself. how this resistance percentage was calculated? And this is for all organisms? One should understand that CLSI guidelnies get changed every year and the antimicrobial susceptibility testing and reporting also changes. does this trend show the prevalence of MDR pathogens over years? No clear information. calculating over all prevalence for ten years doesn’t add much to the community. Not sure whether this might help in surveillance.

Response: Thank you for your questions. We have explained the trend in the method based on your suggestion. In fact, we have explained the overall resistance trend on a year basis. We have shown the plot of resistance pattern (it may be the sum of mono resistant and/or MDR isolates) reported over ten years but not the trend of MDR isolates. 

21. 201. Kindly remove the statement threat as we have more severe threats than UTI S.

Response: corrected accordingly. 

22. 211. there is a huge difference in prevalence from elderly in the previous study and younger age group in the current study. Did the earlier study include all patients from birth? Clarify? If so, one cannot compare. 

Response: The studies we used here for comparison with our result have similar population to our study population. 

23. 213,214,215 what is the relevance of these statements to the current study? 

Response: These have been included to mention the possible rationale that support our result in which UTI was higher in the elderly populations.

24. 236 isolates exceeded 50 percent? Statement or poorly framed and unclear

Response: The statement has been rewritten to make it clear.

25. 244.245 seems to be a loose statement. Is there any reference for local usage of antibiotics statment? 

Response: We have used the term “local usage” to refer the national guidelines for antimicrobial use (we have changed the term to national guidelines for antimicrobial use. Different countries use their own guidelines for antimicrobial use as per their countries’ situations like economic, or distribution of antimicrobial resistance. (Food E. Medicine and Healthcare Administration and Control Authority. Continuing Professional Development (CPD) Guideline for Health Professionals in Ethiopia Addis Ababa: FMoH. 2013. https://www.who.int/selection_medicines/country_lists/Ethiopia_STG_Hosp.pdf)

26. 267 again seems to be a loose statement? Any reference for usage of antibiotics in hiv patients?

Response: It is known that HIV patients are at greater risk of other infections and hence, they are likely to get antibiotics for these infections. Such a high risk of infection and resulting antibiotic treatment may increase the risk of resistance. The WHO has recommended prophylactic prescription of trimethoprim-sulfamethoxazole for HIV infected children. The cited reference itself has described it. 

Reference: (https://doi.org/10.1371/ journal.pone.0243054)

27. 274 earlier it was said that age groups 20- 30 was more affected, this statement is in contradiction with that. Kindly clarify. If its elderly, then define elderly and cite the appropriate analysis for that in results and discussion.

Response: As it is described in the result and table-1, the highest proportion of UTI suspected patients were in the age group 21–30 years (27.1%). However, the rate of infection was highest in the age group 61-70 years (47.4%) followed by patients of the age group above 70 years (44.8%). So, the point we have written in line 274 or in the conclusion is not conflicting. We think you may have seen in the line number -212, “the age group 21-30 was the most affected” but that was not our result rather it was a result reported in Addis Ababa, another part of Ethiopia and used here for comparison. 

B. RESPOSES TO REVIEWER(S) 

1. Reviewer #1: dear authors

There are several lacunae in the methodology part with respect to inclusion and exclusion criteriae. more than one loose statement has been made without citations. quite a lot of errors seen in the discussion section too. kindly provide the details of analysis made for making out the trends in antimicrobial resistance over years. kindly go through the comments and make the necessary revisions

Response: We appreciate you for taking time to thoroughly review this manuscript and offer us your constructive comments and suggestions which will strengthen the quality of our work. We have revised the manuscript and made corrections as necessary based on your comments. We have tried to find and cite appropriate references to every information stated. We also have invited a professional English editor and so, the language and grammatical issues have been curiously amended.

2. Reviewer #2: Comments: 

In method section, sub-section Study area (Line 92), it is stated “We have collected ten years (2010-2019) retrospective data from the microbiology laboratory logbook”. This should describe more detailed e.g., mention how does the author confirm the adequate data and represent for each year period? Moreover, the author should also mention the consistency of practical work in the laboratory for ten years, i.e., method of susceptibility testing, antimicrobial disk used. I think this is very important point as the strength of the manuscript.

Response: we appreciate your concern about the adequacy and representativeness of the data. We have collected patients’ records which were complete record of the variables mentioned in the inclusion criteria within the whole study period. Moreover, representativeness of the collected data should not be the concern here because we have collected all the available data processed each year and contain complete record of the required variables, we did not take a sample.

Regarding the consistency of the laboratory work: The Hospital is one of the biggest pioneering hospitals in the country, Ethiopia. The laboratory performs microbiology culture constantly throughout the year and had no interruption over the study period. We know the laboratory as we are working with the hospital and one of the authors, Mr. Michael Getie, is manager of the microbiology laboratory. The UOGCSH microbiology laboratory had been using disc diffusion method of AST has been used in the laboratory. The antimicrobial agents could have been used up for a brief period until purchased but as the national antimicrobial treatment guideline was not changed long term discontinuation of antimicrobials has not occurred and antimicrobial disk used have been are mentioned in the method section. So, commonly prescribed discs have been used consistently.

3. In method section, sub-section population (Line 94-95), it is stated “all symptomatic patients who visited the hospital from 2010-2019 and had recorded urine culture results were the study population”. Please describe how the author performs the screening of the patient! This is important information to be added in the method section. Moreover, description about elaboration data with microbiology aspect is needed, as appear in Line 129-130: “The data were collected by investigators with quality and completeness checks throughout the collection period, at the end of data collection, and after entry to SPSS for statistical analysis.” Also, need to define the definition for trend of AMR in this study!

Response: As this study is based on a retrospective data, we have retrieved the record for urine cultures and collected the data of patients whose age, sex, urine culture result, and antimicrobial susceptibility results (for those with significant bacteriuria) have fully recorded. So, we have not screened patients. We had been cross-checking and re-checking data on a daily basis for the readability, correct data transcription to the data collection checklist, complete filling of the intended variables 

4. In method section, the author should mention how the percentage calculation for resistance trend among antibiotic tested. What is denominator of the calculation? This information is still missing from the method section.

Response: the percentage of resistance was tested by dividing the resistant isolates to the total number of tested isolates to each antibiotic. The percentage of resistance trend was calculated by dividing the sum of resistant isolates in a year to the sum of isolates tested to each agent in the same year. E.g., the sum of resistant isolates identified in 2019/ number of urinary pathogens isolated tested in 2019. 

5. In the inclusion criterion, I think it seems like “negative” statement or opposite from inclusion, please clarify.

Response: We have included the patients’ data if the following hade ben completely recorded

Age, Sex

Urine culture result (both significant bacteriuria and no significant bacteriuria) 

Isolated uropathogen antimicrobial susceptibility result (for the result with significant bacteriuria)

6. In Line 110 – 121: Does it show microbiology procedure? Please write more clearly and elaborate with whole process of the study.

Response: it shows a precise procedure of the urine culture conducted in the microbiology laboratory, where the data was collected. But we have not gone through these laboratory procedures rather we have collected the result done by performing all these procedures as per the standard laboratory operating procedure and the Clinical Laboratory Standards Institute (for selection of antimicrobials and reporting result as susceptible and resistant)

7. Result:

In line 134, the author mentioned about “UTI suspected patient”; but in population section, it was described “all symptomatic patients” (Line 94-95), which means not suspected patient. Please clarify for this sentence! If the author started with suspected patient, then provide the number of patients being suspected and mentioned how many of this become symptomatic patient with culture-proven UTI.

Response: First sorry for the inconvenience, and in the study area, patients showing signs and symptoms are suspected or presumptive UTI patients and will be supposed to give urine for urine culture for microbiological confirmation. When we suspected patients, we mean patients having signs and symptoms of UTI and all suspected patients were symptomatic patients in this study. so, now we have corrected as “UTI suspected” patients. 

8. In table 2: the percentage is confusing and need to defined properly in method section (see the method comment). There is number of 4441, which indicate the number of total specimens, and number of 1072, which indicate the isolate included based on study criteria. Please explain for this two-different information!

Response: The number 4441 is the total number of collected patients’ data (both with significant bacteriuria and with no significant bacteriuria), while the number 1072 represents the total number of patients with significant bacteriuria or simply confirmed UTI patients) but all 4441 patients were included in this study. The % in the 3rd column of table 2 is the percentage of each species of isolate from the total number of patients’ while the 4th column is the percentage of each species of isolate to the total number of patients with significant bacteriuria or total number or organisms isolated in this study. 

9. Lot of mistypes for species name i.e Prouteus, Klebsella. Author need to check all the typing for species name. Also, consistency for using italic formatting.

Response: We have revised and corrected the misspelled organisms.

10. In table 3 and 4: what does the number in bracket indicate? Please provide clear calculation for this table. What is the denominator for the calculation?

In these tables, the denominators indicate the total number of species of isolates tested against to each antimicrobial agent and in the last column the denominator is the total number of all isolates tested to each antimicrobial agent. The denominator is not constant because number of organisms tested varies. The numerator is the percentage of isolates which were resistant from the total tested organisms to a specific antimicrobial tested in a row. The numbers in bracket of in table 3 were the total number of each species isolate and we have canceled the numbers as it is found in table 2 and to avoid repetition.

11. I think the figure 1 and 2 is main finding of this study and should be attached in the body of manuscript, instead as supplementary figure. the result should more address the trend of bacterial and antimicrobial from 10 years observation, instead of mentioned single speces with resistance profile.

The tables were included in the body of the manuscript, but figures were attached as separate file according to the journal manuscript submission guideline.

Discussion:

I don’t see the discussion related with trend as the main title of the manuscript. Please correlate the finding with the term of Trend as mentioned with title of this study.

Response: We have tried to discuss the trend as stated in the discussion part (line 253-259)

---

## [Decision Letter · Decision Letter 1]

14 Oct 2021

PONE-D-21-22966R1Antimicrobial resistance trend of bacterial uropathogens at the university of Gondar comprehensive specialized hospital, northwest Ethiopia: a 10 years retrospective studyPLOS ONE

Dear Dr. Kasew,

Thank you for submitting your manuscript to PLOS ONE. After careful consideration, we feel that it has merit but does not fully meet PLOS ONE’s publication criteria as it currently stands. Therefore, we invite you to submit a revised version of the manuscript that addresses the points raised during the review process.

The resistance percentage is calculated for overall isolates and the trend shows it has been stable over years and has reduced now. I recommend you to clarify the exact criteria used for defining resistance isolates  and whether it is applicable for all the years of the study, which is unlikely as clsi guidelines change every year.Please pay attention to all reviewer comments and submit as early as your convenience.

We look forward to receiving your revised manuscript.

Kind regards,

Monica Cartelle Gestal, PhD

Academic Editor

PLOS ONE

Reviewers' comments:

Reviewer's Responses to Questions

**Comments to the Author**

1. If the authors have adequately addressed your comments raised in a previous round of review and you feel that this manuscript is now acceptable for publication, you may indicate that here to bypass the “Comments to the Author” section, enter your conflict of interest statement in the “Confidential to Editor” section, and submit your "Accept" recommendation.

Reviewer #1: (No Response)

Reviewer #2: All comments have been addressed

2. Is the manuscript technically sound, and do the data support the conclusions?

Reviewer #1: Partly

Reviewer #2: Yes

3. Has the statistical analysis been performed appropriately and rigorously? 

Reviewer #1: No

Reviewer #2: Yes

4. Have the authors made all data underlying the findings in their manuscript fully available?

Reviewer #1: No

Reviewer #2: Yes

5. Is the manuscript presented in an intelligible fashion and written in standard English?

Reviewer #1: Yes

Reviewer #2: Yes

6. Review Comments to the Author

Reviewer #1: Most of the errors which have been pointed out have been corrected

The calculation of trends for individual species could have been a better parameter rather than calculating for overall species. The trend pattern suggests reduction in the percentage of resistance isolates in 2019 as compared to 2010. This is in contradiction to what is seen world wide and even from the same area reported by other authors. Since the CLSI guidelines change the testing pattern of antibiotics over years it would be better if the authors can state the definition of resistant isolates, like to which antibiotics does it refer to. And is this definition kept same for all the years? In such a case this study shows that there has been no increase in the antimicrobial resistance over 10 years which is really surprising

KINDLY provide more clarification of possible

Reviewer #2: 1. Thank you for addressing all the question adequate and properly.

2. However, table 3 and table 4 are still confusing to follow, especially number and percentage. In table 3, e.g Ampicillin, the total row number was 11 isolate/organism; 4 of 11 was Streptococcus spp. and 7 of 11 refer to Entercoccus spp. The proportion of resistance (percentage) for Streptococcus spp and Enterococcus spp, were 50%, 100%, respectively. In the author explanation was stated: "the percentage of resistance was tested by dividing the resistant isolates to the total number of tested isolates to each antibiotic." Referring to table 2, if the total number of isolate per organism, e.g Streptococcus spp was 26, the percentage calculation of 50% also did not fit. Kindly clarify !

3. My suggestion for point 2: if the number of isolate different according to the tested antibiotic, mentioned e.g: 4/8 (50%) or 7/7 (100%), to make the calculation clear; and put table note to highlight the meaning of the number.

7. PLOS authors have the option to publish the peer review history of their article (what does this mean?). If published, this will include your full peer review and any attached files.

Reviewer #1: **Yes: **DR. Sathyamurthy .P

Reviewer #2: No

---

## [Author Response · Author response to Decision Letter 1]

9 Nov 2021

We would like to thank both reviewers for their valuable comments to see the limitations of our manuscript and having us the chance to improve such limitations. 

Response to reviewer(s)

Academic Editor/Reviewer 1: 

The resistance percentage is calculated for overall isolates and the trend shows it has been stable over years and has reduced now. I recommend you to clarify the exact criteria used for defining resistance isolates and whether it is applicable for all the years of the study, which is unlikely as clsi guidelines change every year.

Response: Thank you for your concern. 

The criteria, that has been used to interpret the antimicrobial resistance by the laboratory in which our study was done, is the CLSI guideline. In fact, the CLSI is updated on a yearly basis and the laboratory uses the newly updated versions of CLSI each year. So, the results were reported as susceptible or resistant to the antimicrobial agents used as per the CLSI’s criteria and still no other criteria have been used in the laboratory. Up on release of new versions the CLSI the laboratory gets new versions timely each year and reports as per the new guideline. We are confident that the results are reported following the CLSI guideline and will not deviate the CLSI. E.g., in 2010 antimicrobial susceptibility test (AST) results were interpreted and reported based on 2010’s CLSI AST reporting criteria and that of 2019 results were reported following 2019’s version of CLSI. The number of organisms tested, the number and class of antimicrobial agents used could have a role in the slightly reducing resistance trend. The antimicrobial stewardship program is currently being implemented in the university of Gondar comprehensive specialized hospital and it might also be bringing the intended outcomes in reducing the antimicrobial resistance.

Reviewer #1: 

1. Most of the errors which have been pointed out have been corrected the calculation of trends for individual species could have been a better parameter rather than calculating for overall species. The trend pattern suggests reduction in the percentage of resistance isolates in 2019 as compared to 2010. This is in contradiction to what is seen worldwide and even from the same area reported by other authors. Since the CLSI guidelines change the testing pattern of antibiotics over years it would be better if the authors can state the definition of resistant isolates, like to which antibiotics does it refer to. And is this definition kept same for all the years? In such a case this study shows that there has been no increase in the antimicrobial resistance over 10 years which is really surprising.

KINDLY provide more clarification of possible

Response: first we want thank you for considering our previous response. We also acknowledge your attention and critical review to make our work scientifically valuable. 

2. The calculation of trends for individual species could have been a better parameter rather than calculating for overall species

Response: 

This study was a retrospective study done by collecting a ten years laboratory data. The laboratory the antimicrobial discs used for antimicrobial susceptibility test (AST) were not consistently used throughout 10 years due to supply interruptions of the discs. In the absence of full stock of antimicrobial discs, the laboratory uses only few discs which are available at hand. The other is the frequency of species isolated and tested shows huge difference from too few (e.g., Serratia species = 4, Providencia species =5) to hundreds (e.g., E. coli =404) which makes the trend by species difficult to compare and conclude. So, calculating the species wise trend by species will not be important for conclusion. That is why we were not interested in analysis of species wise trend of resistance, but we have mentioned this as limitation. 

3. The trend pattern suggests reduction in the percentage of resistance isolates in 2019 as compared to 2010. This is in contradiction to what is seen worldwide and even from the same area reported by other authors.

Response: 

Yes, the trend was inconsistent and it seems that the percentage of resistance reported in 2010 was a little higher than the percentage of resistance reported in 2019. Multiple factors could have contributed for this seemingly contradiction. To mention some, the number of organisms tested, the number and class of antimicrobial agents used could have a role in the slightly reducing resistance trend. The tendency to replace antimicrobial agents with more potent discs could have reduced or maintain the resistance rate at steady level. For instance, there were higher frequency of ampicillin, amoxicillin use in 2010, 2011 and reduced the frequency of using these agents in recent years. Conversely, nitrofurantoin, which is a potent antibiotic used in the treatment of urinary tract infection, has not been used until 2014 and become increasingly used in the recent years. Moreover, 90.53% of meropenem disks were used for AST in the last two years (2018-2019). Amikacin has been used since 2017 and kanamycin has been used in 2018& 2019, which indicates drugs used for AST are changing. So, this phenomenon could have reduced the pooled percentage of resistance. The antimicrobial stewardship program is currently being implemented in the university of Gondar comprehensive specialized hospital and it might also be bringing the intended outcomes in reducing the antimicrobial resistance.

4. Since the CLSI guidelines change the testing pattern of antibiotics over years it would be better if the authors can state the definition of resistant isolates, like to which antibiotics does it refer to. And is this definition kept same for all the years? In such a case this study shows that there has been no increase in the antimicrobial resistance over 10 years which is really surprising. 

Response: 

We have collected data which have already been processed and recorded by the laboratory. So, we have collected antimicrobial susceptibility test (AST) results which have been reported as susceptible or resistant in each year and the laboratory uses each year’s update of the CLSI for interpretation of AST results as susceptible or resistant. E.g., in 2010 antimicrobial susceptibility test (AST) results were interpreted and reported based on 2010’s CLSI AST reporting criteria and that of 2019 results were reported following 2019’s version of CLSI. 

Reviewer #2: 

1. Thank you for addressing all the question adequate and properly. 

2. However, table 3 and table 4 are still confusing to follow, especially number and percentage. In table 3, e.g Ampicillin, the total row number was 11 isolate/organism; 4 of 11 was Streptococcus spp. and 7 of 11 refer to Entercoccus spp. The proportion of resistance (percentage) for Streptococcus spp and Enterococcus spp, were 50%, 100%, respectively. In the author explanation was stated: "the percentage of resistance was tested by dividing the resistant isolates to the total number of tested isolates to each antibiotic." Referring to table 2, if the total number of isolates per organism, e.g Streptococcus spp was 26, the percentage calculation of 50% also did not fit. Kindly clarify! 3. My suggestion for point 2: if the number of isolate different according to the tested antibiotic, mentioned e.g: 4/8 (50%) or 7/7 (100%), to make the calculation clear; and put table note to highlight the meaning of the number. 

Response: Thank you dear reviewer.

We have stated in our previous revision of the comments percentage of resistance was determined by dividing number of resistant isolates to the number of isolates. It is important to note that all isolates could have not been tested to each antibiotic. For instance, the total number of Streptococcus spp isolated in this study were 26, among these isolates only were 8 were tested against ampicillin and 50% of which were resistant to ampicillin. Why all isolated organisms were not tested could be due to availability of antibiotic discs, probably due to stock out or selection of priority agents. 

We intended to minimize space if it is confusing, to avoid such a confusion we have corrected tables as per your comment.

---

## [Decision Letter · Decision Letter 2]

16 Dec 2021

PONE-D-21-22966R2Antimicrobial resistance trend of bacterial uropathogens at the university of Gondar comprehensive specialized hospital, northwest Ethiopia: a 10 years retrospective studyPLOS ONE

Dear Dr. Kasew,

Thank you for submitting your manuscript to PLOS ONE. After careful consideration, we feel that it has merit but does not fully meet PLOS ONE’s publication criteria as it currently stands. Therefore, we invite you to submit a revised version of the manuscript that addresses the points raised during the review process.

 Thanks for this revised version of the manuscript. It has significantly improve. There are however some suggestions that need to be addressed before moving forward, as one of the reviewers has major concerns with the manuscript in its current form. Please carefully address all the comments here provided.Looking forward to seeing the next submission

We look forward to receiving your revised manuscript.

Kind regards,

Monica Cartelle Gestal, PhD

Academic Editor

PLOS ONE

Reviewers' comments:

Reviewer's Responses to Questions

**Comments to the Author**

1. If the authors have adequately addressed your comments raised in a previous round of review and you feel that this manuscript is now acceptable for publication, you may indicate that here to bypass the “Comments to the Author” section, enter your conflict of interest statement in the “Confidential to Editor” section, and submit your "Accept" recommendation.

Reviewer #2: All comments have been addressed

Reviewer #3: All comments have been addressed

2. Is the manuscript technically sound, and do the data support the conclusions?

Reviewer #2: Yes

Reviewer #3: Partly

3. Has the statistical analysis been performed appropriately and rigorously? 

Reviewer #2: N/A

Reviewer #3: No

4. Have the authors made all data underlying the findings in their manuscript fully available?

Reviewer #2: Yes

Reviewer #3: Yes

5. Is the manuscript presented in an intelligible fashion and written in standard English?

Reviewer #2: Yes

Reviewer #3: Yes

6. Review Comments to the Author

Reviewer #2: Thank you for addressing the question properly. The last comment for reviewer2 has been answered properly, however refer to other reviewer comment to the different trend pattern, my suggestion is author can explain the outbreak occurrence and/or surveillance method approach which may be useful as important point for discussion.

Reviewer #3: Manuscript number PONE-D-21-22966R2

The authors tried to address most comments or concerns raised by the previous reviewers; however, still the authors should address the following comments to improve the quality of the manuscript:

Methods

The authors mentioned that the culture media and biochemical tests used. Since the laboratory logbook has been documented only patients’ age, sex, address, urine culture results and antimicrobial susceptibility test (AST) results and over the years the hospital may use different culture media and biochemical tests, so how did you trace the type of the media and biochemical tests used?

The authors are advised to perform PCR-based detection of the antimicrobial resistance genes in multidrug-resistant strains if applicable. (On the other hand, address this point to the study limitations).

Population: I think the study populations are not the patients that visited the hospital over the 10-year period rather it could be the urine culture results records. Please kindly explain about it.

Results:

One of the main of the study was to show the drug resistance patterns of the isolates over the specified time. However, the study lacks to reveal antibiotics/ class of antibiotics have a trend of increasing resistance over 10 years period. So, kindly request the authors to show the trend of resistance for each class of antibiotics on a yearly basis.

Still the calculation of trend over the years is not clear. It could be a better parameter and more explanatory if the calculation is based on the isolated species and a yearly base rather than considering the overall isolates.

Antimicrobial resistance (AMR)

It could be better the study showed that MDR, XDR and PDR isolates from each class of the antimicrobials and please illustrate each class of antibiotics used and resistance patterns.

In figure 2, it showed the overall resistance species; it could be better if it depicts the resistance species and its trend over the years.

There are some calculations errors. For example, table three, the first row the denominators should be 15 not 13. Please curiously address such issues.

In this table, I recommend to make one last row, label total isolates, and work out the total resistance isolates for each species. For example, for CoNS, total tested isolates 172, resistance isolates 109 and rate of resistant was 63.37%.

The same is true for table 4. This table somewhat congested and please make it clear.

When the authors say LFGNR and NLFGNR, what it means? The authors mention LFGNR including E. coli, Klebsiella species, and Citrobacter and Enterobacter species. So, what it stands? The same is true for NLFGNR. Unless you grouped together or indicated other that the stated species, please avoid such jargon terminologies.

Discussions:

The authors are advised to illustrate the real impact of their findings in the managements of UTI in hospital as well as in the global community at large without repetition of results.

Minor issues:

LINE 51: More than 70% of isolates were resistant to…. Which isolates GPB or GNB?

LINE 69: 9) What is it number?

LINE88: …. retrospective cross-sectional…. What kind of study design?

LINE 113-121: Where did the authors find the information/data? Did the data collected at the time of the tested performed? Please kindly explain where the source is.

7. PLOS authors have the option to publish the peer review history of their article (what does this mean?). If published, this will include your full peer review and any attached files.

Reviewer #2: No

Reviewer #3: No

---

## [Author Response · Author response to Decision Letter 2]

9 Jan 2022

Response to Reviewers

First we would like to acknowledge the reviewers for the valuable comments and questions so that our manuscript will be mature scientific.

Response to Reviewer-2

Reviewer #2: Thank you for addressing the question properly. The last comment for reviewer2 has been answered properly, however refer to other reviewer comment to the different trend pattern, my suggestion is author can explain the outbreak occurrence and/or surveillance method approach which may be useful as important point for discussion.

Response: Thank you for considering our previous responses and your current suggestion! But this study was done to determine the burden of urinary tract infection (UTI) over a long (ten years) period retrospectively but outbreak was not seen nor surveillance of that outbreak were used to investigate occurrence of the disease. So, we have determined the prevalence of infection and resistance pattern of isolates retrospectively.

Response to Reviewer-3

I must acknowledge you for your comments and suggestions to improve the quality of our manuscript.

Methods 

1. The authors mentioned that the culture media and biochemical tests used. Since the laboratory logbook has been documented only patients’ age, sex, address, urine culture results and antimicrobial susceptibility test (AST) results and over the years the hospital may use different culture media and biochemical tests, so how did you trace the type of the media and biochemical tests used?

Response: Thank you for the concern: The laboratory has standard operating procedure (SPO) in which the culture media used in the University of Gondar comprehensive specialized hospital laboratory and preparation of culture media is documented there and we have used that document as our source.

2. The authors are advised to perform PCR-based detection of the antimicrobial resistance genes in multidrug-resistant strains if applicable. (On the other hand, address this point to the study limitations).

Response: The study was retrospective and we directly took the record of the data. Even the laboratory has no molecular detection methods. 

We have accepted the comment and included in the limitation 

3. Population: I think the study populations are not the patients that visited the hospital over 10-year period rather it could be the urine culture results records. Please kindly explain about it.

Response: We have accepted your comment and corrected

Results:

4. One of the main of the study was to show the drug resistance patterns of the isolates over the specified time. However, the study lacks to reveal antibiotics/ class of antibiotics have a trend of increasing resistance over 10 years period. So, kindly request the authors to show the trend of resistance for each class of antibiotics on a yearly basis.

Response: It is done in a class of antimicrobial agents on a yearly basis in table 5.

5. Still the calculation of trend over the years is not clear. It could be a better parameter and more explanatory if the calculation is based on the isolated species and a yearly base rather than considering the overall isolates.

Response: the resistance trend in species was not done. Because, as it can be seen in table 5, antimicrobial agents were inconsistently used. E.g., carbapenem drugs were used only in the last two years. Isolates tested against penicillin (144) were not tested against aminoglycosides (100) and even among these isolates one could be tested to one but not to the other. This limited our analysis of trend on a species basis. Hence, it is more informative to analyze on a yearly basis and made a pooled resistance rate on a yearly basis can be conclusive.

Antimicrobial resistance (AMR)

6. It could be better the study showed that MDR, XDR and PDR isolates from each class of the antimicrobials and please illustrate each classes of antibiotics used and resistance patterns. 

Response: Thank you!

To minimize congested nature of table, we have canceled the column of antimicrobial classes in other tables and mentioned classes of antimicrobial agents in Table 5.

As there are antimicrobial classes (such as Glycylcyclines, Monobactams…), that were not available in the laboratory and AST was not done to these agents, it is difficult to report XDR and PDR isolates. The definition is “MDR is defined as non-susceptibility to at least one agent in three or more antimicrobial categories. XDR is defined as non-susceptibility to at least one agent in all but two or fewer antimicrobial categories (i.e., bacterial isolates remain susceptible to only one or two categories). PDR is defined as non-susceptibility to all agents in all antimicrobial categories (i.e., no agents tested as susceptible for that organism)”. Hence, according to the definition we have only MDR isolates. The research is a retrospective study and we have analyzed and reported what had been done and recorded. 

(Reference: Magiorakos AP, Srinivasan A, Carey RB, Carmeli Y, Falagas ME, Giske CG, Harbarth S, Hindler JF, Kahlmeter G, Olsson-Liljequist B, Paterson DL. Multidrug-resistant, extensively drug-resistant and pandrug-resistant bacteria: an international expert proposal for interim standard definitions for acquired resistance. Clinical microbiology and infection. 2012 Mar 1;18(3):268-81. https://doi.org/10.1111/j.1469-0691.2011.03570.x)

7. In figure 2, it showed the overall resistance species; it could be better if it depicts the resistance species and its trend over the years.

Response: The resistant species were mentioned in table -3 and table 4, but the trend of resistance to each species was not done because antimicrobial agents were not consistently used to test each isolates and over years. So, we showing the trend of resistance over the years was done. Seen in No 5.

8. There are some calculations errors. For example, table three, the first row the denominators should be 15 not 13. Please curiously address such issues. 

Response: tables were revised and corrected as per the comment

9. In this table, I recommend to make one last row, label total isolates, and work out the total resistance isolates for each species. For example, for CONS, total tested isolates 172, resistance isolates 109 and rate of resistant was 63.37%

Response: It can be seen in the table different denominators indicating different number of isolates were tested against each antimicrobial agent. So, different antibiotics are difficult to add up. For example, only one isolate of CONs was tested against Cefoxitin while 17 were tested against Trimethoprim-sulfamethoxazole (SXT) and that one isolate could or could not be tested against SXT. 

10. The same is true for table 4. This table somewhat congested and please make it clear.

Response: we understand that the table is congested because of the number of drugs used and species of isolates were diverse. It could be good to cancel the denominators, it was submitted that way in our 1st submission and changed to the current form with comment.

11. When the authors say LFGNR and NLFGNR, what it means? The authors mention LFGNR including E. coli, Klebsiella species, and Citrobacter and Enterobacter species. So, what it stands? The same is true for NLFGNR. Unless you grouped together or indicated other that the stated species, please avoid such jargon terminologies.

Response: LFGNR-Lactose fermenting Gram negative rods, NLFGNR- None lactose fermenting Gram negative rods.

There were records in which the genus and species isolated was not specified and were reported as LFGNR and NLFGNR. This is a retrospective study done by copying the record from the laboratory. Hence it is difficult to add to the specified species and we put these isolates in separate columns.

Discussions:

12. The authors are advised to illustrate the real impact of their findings in the managements of UTI in hospital as well as in the global community at large without repetition of results.

Response: 

This study figures out the burden of UTI and the rate of antimicrobial resistance to the available antimicrobials. So, the hospitals in the locality can understand which antibiotics can be used in imperial treatment and which agent should not be prescribed. The global community can also get the figure in the region.

The expected impact of the study was stated in the introduction part line 77-81, line 285-293 as well as in the conclusion (line 303-307). 

Minor issues: 

LINE 51: More than 70% of isolates were resistant to…. Which isolates GPB or GNB?

Response: Done as per the comment. It includes both gram positive and gram-negative isolates. 

LINE 69: 9) What is it number? 

Response: It was a typographical error and now corrected.

LINE88: …. retrospective cross-sectional…. What kind of study design?

Response: the study design is a retrospective study design and cross sectional was added to indicate the data collected period. it is corrected as retrospective study.

LINE 113-121: Where did the authors find the information/data? Did the data collected at the time of the tested performed? Please kindly explain where the source is.

Response: The data were collected from the University of Gondar comprehensive specialized hospital microbiology laboratory record book.

---

## [Decision Letter · Decision Letter 3]

30 Mar 2022

Antimicrobial resistance trend of bacterial uropathogens at the university of Gondar comprehensive specialized hospital, northwest Ethiopia: a 10 years retrospective study

PONE-D-21-22966R3

Dear Dr. Kasew,

We’re pleased to inform you that your manuscript has been judged scientifically suitable for publication and will be formally accepted for publication once it meets all outstanding technical requirements.

Kind regards,

Simon Clegg, PhD

Academic Editor

PLOS ONE

Additional Editor Comments:

Many thanks for resubmitting your manuscript to PLOS One

As you have addressed all the comments and the manuscript reads well, I have recommended it for publication

You should hear from the Editorial Office shortly.

It was a pleasure working with you and I wish you the best of luck for your future research

Hope you are keeping safe and well in these difficult times

Thanks

Simon

Reviewers' comments:

Reviewer's Responses to Questions

**Comments to the Author**

1. If the authors have adequately addressed your comments raised in a previous round of review and you feel that this manuscript is now acceptable for publication, you may indicate that here to bypass the “Comments to the Author” section, enter your conflict of interest statement in the “Confidential to Editor” section, and submit your "Accept" recommendation.

Reviewer #2: All comments have been addressed

Reviewer #3: All comments have been addressed

2. Is the manuscript technically sound, and do the data support the conclusions?

Reviewer #2: Yes

Reviewer #3: Partly

3. Has the statistical analysis been performed appropriately and rigorously? 

Reviewer #2: N/A

Reviewer #3: Yes

4. Have the authors made all data underlying the findings in their manuscript fully available?

Reviewer #2: Yes

Reviewer #3: Yes

5. Is the manuscript presented in an intelligible fashion and written in standard English?

Reviewer #2: Yes

Reviewer #3: Yes

6. Review Comments to the Author

Reviewer #2: Thank you for addressing the previous comment clearly, at this moment I don't have specific question referring to the manuscript, except:

1. Consistency of percentage calculation in Table 1, e.g in Frequency column, the percentage was calculated between male (45.2%, 2006 of 4441) to female (54.8%, 2435 of 4441) (column as total column of denominator); but percentage between UTI positive and negative was using total row, shown in the table for male, positive UTI = 24.6%, 494 of 2006 an negative UTI = 75.4%, 1512 of 2006. My suggestion, it will be better to use the total column as the denominator of the percentage, or please give some footnote that declare the different denominator of percentage calculation.

2. In table 5, please consistent with antibiotic class/group name, e.g SXT = folate pathway inhibitor. The antibiotic group/class can be referred to reference no.18 (Magiorakos et. al.)

Reviewer #3: Overall, the author tried to address all questions and comments.

Thank you for addressing the question properly

7. PLOS authors have the option to publish the peer review history of their article (what does this mean?). If published, this will include your full peer review and any attached files.

Reviewer #2: No

---

## [Editor Report · Acceptance letter]

1 Apr 2022

PONE-D-21-22966R3 

Antimicrobial resistance trend of bacterial uropathogens at the university of Gondar comprehensive specialized hospital, northwest Ethiopia: a 10 years retrospective study 

Dear Dr. Kasew:

I'm pleased to inform you that your manuscript has been deemed suitable for publication in PLOS ONE. Congratulations! Your manuscript is now with our production department. 

Kind regards, 

on behalf of

Dr. Simon Clegg 

Academic Editor

PLOS ONE